# A novel nutritional supplement containing amino acids and chromium decreases postprandial glucose response in a randomized, double-blind, placebo-controlled study

Elin Östman[1][¤]*, Azat Samigullin[2], Lovisa Heyman-Lindén[3], Kristina Andersson[3,4], Inger Björck[1], Rickard Öste[1,3], Per M. Humpert[2,5,6]

1 Food for Health Science Center, Lund University, Lund, Sweden, 2 StarScience GmbH, Heidelberg, Germany, 3 Aventure AB, Lund, Sweden, 4 Department of Experimental Medical Science, Lund University, Lund, Sweden, 5 Stoffwechselzentrum Rhein Pfalz, Mannheim, Germany, 6 Department of Internal Medicine, University of Heidelberg, Heidelberg, Germany

¤ Current address: Good Idea, Inc., Larkspur, California, United States of America
* elin.ostman@goodideadrinks.com

**Citation:** Östman E, Samigullin A, Heyman-Lindén L, Andersson K, Björck I, Öste R, et al. (2020) A novel nutritional supplement containing amino acids and chromium decreases postprandial glucose response in a randomized, double-blind, placebo-controlled study. PLoS ONE 15(6): e0234237. https://doi.org/10.1371/journal.pone.0234237

## Abstract

High postprandial blood glucose levels are associated with increased mortality, cardiovascular events and development of diabetes in the general population. Interventions targeting postprandial glucose have been shown to prevent both cardiovascular events and diabetes. This study evaluates the efficacy and safety of a novel nutritional supplement targeting postprandial glucose excursions in non-diabetic adults. Sixty overweight healthy male and female participants were recruited at two centers and randomized in a double-blind, placebo-controlled, crossover design. The supplement, a water-based drink containing 2.6g of amino acids (L-Leucine, L-Threonine, L-Lysine Monohydrochloride, L-Isoleucine, L-Valine) and 250 mcg of chromium picolinate, was consumed with a standardized carbohydrate-rich meal. The primary endpoint was the incremental area under the curve (iAUC) for venous blood glucose from 0 to 120 minutes. Secondary endpoints included glucose iAUC $_{0-180\ minutes}$ and the maximum glucose concentration ($C_{max}$), for both venous and capillary blood glucose. In the intention-to-treat-analysis (n = 60) the supplement resulted in a decreased venous blood glucose iAUC$_{0-120min}$ compared to placebo, mean (SE) of 68.7 (6.6) versus 52.2 (6.8) respectively, a difference of -16.5 mmol/L•min (95% CI -3.1 to -30.0, $p$ = 0.017). The $C_{max}$ for venous blood glucose for the supplement and placebo were 6.45 (0.12) versus 6.10 (<0.12), respectively, a difference of -0.35 mmol/L (95% CI -0.17 to -0.53, $p$<0.001). In the per protocol-analysis (n = 48), the supplement resulted in a decreased $C_{max}$ compared to placebo from 6.42 (0.14) to 6.12 (0.14), a difference of -0.29 mmol/L (95% CI -0.12 to -0.47, $p$ = 0.002). No significant differences in capillary blood glucose were found, as measured by regular bed-side glucometers. The nutritional supplement drink containing amino acids and chromium improves the postprandial glucose homeostasis in overweight adults without diabetes. Future studies should clarify, whether regular consumption of the

**Data Availability Statement:** All relevant data are within the paper and its Supporting Information files.

**Funding:** The funder provided support in the form of salaries [EÖ, LHL, KA and RÖ] and consulting fees [starScience GmbH] for authors but did not have any additional role in the study design, data collection and analysis, decision to publish, or preparation of the manuscript. The specific roles of these authors are articulated in the 'author contributions' section.

**Competing interests:** I have read the journal's policy and the authors of this manuscript have the following competing interests: EÖ, IB and RÖ are inventors of a patent family describing the supplement studied. EÖ and IB jointly own the right to the patent and Aventure AB/Double Good AB (RÖ) owns a license to use the patent. EÖ is an employee of Good Idea, Inc since August 2017. starScience GmbH (AS, PMH) have received funding for other studies by Aventure AB/Double Good AB. PH holds shares of Double Good AB. KA and LHL are employees of Aventure AB, the parent company of Double Good AB and Good Idea, Inc. The commercial affiliations of the authors do not alter our adherence to PLOS ONE policies on sharing data and materials.

supplement improves markers of disease or could play a role in a diet aiming at preventing the development of diabetes.

## Introduction

High postprandial glucose levels after an oral glucose load are associated with the development of type 2 diabetes (T2D), cardiovascular disease (CVD) as well as increased mortality even in absence of a pre-diabetic condition such as impaired fasting glucose (IFG) [1–8]. A causal relationship between high postprandial glucose levels and T2D or CVD appears likely since interventions targeting postprandial glycemia have been shown to prevent these clinical conditions. This has explicitly been shown for pharmacologic interventions with acarbose [9–11] and is robustly supported by studies on low glycemic index (GI) diets [12–17] as well as experimental data describing possible underlying mechanisms [18–21].

Prevention of T2D and CVD are highly relevant and urgent public health issues since rates of obesity and overweight as well as the age of populations increase [1]. At the same time carbohydrate-rich foods with a high GI and a high glycemic load (GL), which became widespread throughout the 20th century paralleled by an increasing prevalence of obesity and T2D [22], are very popular and commonly available to the consumer [23]. These foods bear a high potential of causing pronounced glucose excursions and insulin spikes, which in turn might contribute to the development of T2D and CVD.

The aim of this study was to show efficacy and safety of a novel nutritional supplement drink (hereinafter called "the supplement")—a blend of five specific amino acids (5AA) and chromium picolinate (CrPic) in water, in a two center, double-blinded RCT. The 5AA were selected on basis of their rapid appearance in the blood stream after intake of whey protein [24] and chromium picolinate for its potential effect on insulin sensitivity [25]. The supplement has been developed to lower postprandial glycemia when consumed with a carbohydrate-rich meal and showed promising results in published [26] and unpublished pilot studies.

## Methods

### Study design

The randomized, double-blind, placebo-controlled, crossover study was performed by KGK Science Inc. at two centers (London ON, Canada and Orlando FL, US). Participants attended two single day periods separated by a 7-day washout period.

### Ethics

This trial was registered at clinicaltrials.gov as NCT03152682. The registration was done with some delay with respect to patient recruitment, due to lack of clarity who was in charge of the submission. The authors confirm that all ongoing and related trials for this intervention are registered. The investigational product was reviewed by Health Canada and classified as a food. Issuance of a permit to conduct this clinical trial in Canada was therefore not required by Health Canada. Similarly, an Investigational New Drug permit was not required for conduct of the study in the USA. This study was approved for conduct at both the Canadian and the USA sites by the Institutional Review Board (IRB Services, Aurora, Ontario) on March 14, 2017. It was conducted in accordance with the ethical principles that have their origins in the Declaration of Helsinki and its subsequent amendments. Amendments to the protocol were

approved on June 14, 2017 and July 27, 2017. Written informed consent was obtained from all participants. The investigational products were produced in the USA and no permit for the import of the investigational product into Canada was required.

## Participants

Individuals were recruited from Southwestern Ontario, Canada and Orlando, Florida, using the KGK Science Inc.'s internal participant database along with local electronic and physical advertisement devoid of gender or racial bias. Participants were required to be between 18 and 50 years of age, have a BMI between 25–29.9 kg/m$^2$ and be in a good physical and mental condition according to their own perception (on a five point scale from 'poor' to 'excellent'), medical history and laboratory results (full list of inclusion and exclusion criteria in S1 Table).

## Test product and placebo

The supplement was a lightly carbonated drink containing a proprietary blend with molar ratios presented in [26] and a total of 2.6 g 5AA (L-Leucine, L-Threonine, L-Lysine Monohydrochloride, L-Isoleucine and L-Valine) as well as 250 mcg CrPic (Good Idea®, Good Idea, Inc., Larkspur, CA). Besides water and the active ingredients, the test product also contained citric acid, lemon natural flavor, sodium benzoate and potassium sorbate. The placebo product was identical to the test product with respect to all ingredients, but 5AA and CrPic. The investigational product and placebo were sealed in identically appearing bottles and matched in regard to color, taste and texture.

## Intervention

After an overnight fast of 12h, the participants reported to the study facility in the morning. Upon confirmation of participant status and eligibility criteria, an intravenous cannula for repeated blood sampling was placed into the antecubital vein.

Participants started by consuming 175 ml of the supplement or placebo on an empty stomach (time 0), and 3 min later they began with the ingestion of a standardized breakfast meal, alternating between eating and drinking. Three portions of 50 ml each of the supplement or placebo were required to be consumed at min 3, 7 and 11 and the breakfast meal was to be completed by min 14. At the 14 min mark, the final 30 ml of test product or placebo were ingested. The total amount of supplement or placebo was 355 ml. The test meal consisted of white wheat bread (110 g without crust), butter (10 g) and ham (38 g). The calorie and macronutrient composition of the standardized breakfast meal at the London clinical site was: 377 kcal, 47.7g carbohydrates, 14.9 g proteins and 14.1 g fats. The corresponding figures for the Orlando site were: 350 kcal, 45.3 g carbohydrates, 13.1 g proteins and 13 g fats.

Blood samples were collected at 0, 15, 30, 45, 60, 90, 120 and 180 min for glucose (venous and capillary) and insulin (venous) analysis. Capillary blood glucose was measured by the CONTOUR® monitoring system (London site) and the OneTouch® Ultra 2 monitoring system (Orlando site). Quantitative determination of glucose in human serum was conducted via the enzymatic method with hexokinase utilizing the Roche Cobas e701 Analyzer (sensitivity limit 0.5 mmol/L, intra-assay coefficient of variation (CV) 2.7% at 3.1 mmol/L, and 1.4% at 19.8 mmol/L). Quantitative determination of insulin in human serum was conducted via electrochemiluminescence immunoassay utilizing the Roche Cobas e602 Analyzer (sensitivity limit 4 pmol/L, intra-assay CV 7.5% at 106 pmol/L, and 4.5% at 1 357 pmol/L).

Safety endpoints were analyzed in blood drawn at screening and both investigational visits (before starting the actual intervention and associated blood sampling) and included: blood count (hemoglobin, hematocrit, platelet count, red blood cell count, red cell indices, red cell

distribution width, white blood cell count, and differentials (neutrophils, lymphocytes, mono-cytes, eosinophils, basophils)), liver function (alanine transaminase, aspartate transaminase, bilirubin), and kidney function (creatinine, electrolytes (Na, K, Cl)). Besides the blood parameters, also blood pressure and heart rate were monitored. Urine pregnancy tests (Biostrip HCG, Innovatek Medical Inc.) were conducted at both sites for participants of childbearing age during the screening and baseline visits.

## Outcome measures

The pre-defined primary outcome variable was 120-minute incremental area under the curve ($iAUC_{0-120min}$) for venous serum glucose. Time zero (0) indicates the time when the participant started eating the standardized meal and then repeated blood samples were taken until the last one at 120 min post meal start. The iAUC is the area under the curve, above the baseline levels. It corrects for variations in fasting plasma glucose levels within and between individuals and is thus optimal for comparing the effect of a food on blood glucose levels of individuals independently of their fasting plasma glucose. The iAUC is used, among others, for the calculation of the glycemic index [27]. Secondary outcome variables for venous sampling were: glucose $iAUC_{0-180min}$; serum insulin $iAUC_{0-120min}$ and $iAUC_{0-180min}$; peak glucose value in 120 minutes ($C_{max\ 0-120min}$) and 180 minutes ($C_{max\ 0-180min}$); time to peak glucose for 120 minutes ($T_{max\ 0-120min}$) and 180 minutes ($T_{max\ 0-180min}$); serum insulin $C_{max\ 0-120min}$ and $C_{max\ 0-180min}$. Secondary outcome variables for capillary measurements were glucose $iAUC_{0-120min}$ and $iAUC_{0-180min}$; $C_{max\ 0-120min}$ and $C_{max\ 0-180min}$; $T_{max\ 0-120min}$ and $T_{max\ 0-180min}$, as well as eating patterns as assessed by a 3-day food record for the weeks prior to days 0 and 8. The difference in each secondary outcome between supplement group and placebo group was tested separately by applying the method described in the Statistical analysis section.

## Sample size calculation

A sample size of 30 participants per center (total of 60 participants) was calculated based on an expected iAUC of 66.09 mmol•min/L for the placebo and a pooled standard deviation of 61.50 mmol•min/L from a study by Lustig et al [28]. A mean difference of 36.8 mmol/L/min between the investigational product group and the placebo group is expected for the iAUC for each center, assuming a two-sided test with alpha equal to 5%, 80% power and a 20% drop-out rate from enrollment to final, post-baseline measurement.

## Randomization

Randomization numbers were assigned by a blinded investigator per the order of a randomization list generated by www.randomization.com and allocated to their sequence in a decentralized manner in a 1:1 ratio, with a separate randomization list for each site. An un-blinded associate that was not involved in any data capture or study assessments labeled the investigational product. A randomization schedule was created and provided to the investigator indicating the order of randomization. Investigators, other site personnel and participants were blinded to the products.

## Statistical analysis

All analyses were performed on the pooled data of the two centers. The following analytical populations were defined for the study: Safety population–all participants who received either product and on whom any post-randomization safety information was available; Intention-to-treat (ITT) population–all participants who received either product and on whom any post-

randomization efficacy information was available; Per Protocol (PP) population–all participants who consumed 100% of the investigational products, did not have any major protocol violations, and completed all study visits and procedures connected with measurement of the primary variable.

The trapezoid rule was used for calculating $iAUC_{0-120min}$ and $iAUC_{0-180min}$. The trapezoidal rule is a numerical integration method used to approximate the area under a curve. It is widely used to calculate the area under pharmacokinetic curves [29, 30]. Although simple, the trapezoidal rule is a reliable method for calculating iAUC, which has been shown to be more strongly correlated to glycemic response than total AUC [31, 32]. For the iAUC calculations, invalid data points in the ITT and PP analysis were handled by simple imputation methods–averaging values directly adjacent to the missing data point or using a participant's corresponding value from the other study period (pre-ingestion). It was possible to calculate the primary endpoint for 57 participants. Calculations for all other outcome variables were performed on the original data set without imputation. The between group changes from pre-ingestion were analyzed by a mixed model repeated measures analysis of variance. The model included subject as a random effect, with fixed effects of group, sequence, and visit. The p-values for each change was derived by a linear contrast statement of this model. P-values $\leq 0.05$ were considered statistically significant. All statistical analysis was completed using the R Statistical Software Package Version 3.2.2 (R Core Team, 2015) and SAS version 9.3 (Cary, North Carolina) for Microsoft Windows.

## Results

The participant recruitment began in March 2017 and the study was completed by August 2017. In order to recruit 60 men and women in equivalent numbers and according to pre-defined inclusion and exclusion criteria, a total of 110 candidates needed to be screened. The participant flow is outlined in Fig 1.

The ITT population included 60 participants (32 female, 28 male) from which 12 were excluded, with a resulting PP population of 48 participants (23 female, 25 male). Baseline characteristics are presented in Table 1. Out of the 12 excluded participants, seven were from the supplement → placebo sequence and five from the placebo → supplement sequence. The reasons for the exclusions were: starting antibiotic treatment (n = 1), delayed or missing blood samples (n = 3), non-adherence to the protocol (n = 2), fainting (n = 1) and wish to terminate the participation before second trial day (n = 5).

The glucose and insulin responses are presented in Fig 2 and Table 2.

In the ITT population, the supplement led to a 24% reduction in venous serum glucose $iAUC_{0-120\ min}$ (p = 0.03) as the primary outcome measure as well as a 5.4% reduction in $C_{max\ 0-120}$ (p<0.01) as compared to placebo. There were significant differences in serum glucose between placebo and the supplement at 30, 45, 60, 90 and 120 minutes (reduction by 4%, 3%, 5%, 6% and 4% respectively). No significant differences in the iAUCs for insulin, $T_{max}$ or iAUC for capillary measurements were observed.

In the PP population a 4.5% reduction in $C_{max\ 0-120}$ (p<0.01) could be documented. There were significant reductions in serum glucose by the supplement compared with placebo at 30, 45 and 60 minutes (-4%, -4% and -5% respectively). No significant differences in the iAUCs for venous serum glucose or insulin as well as $T_{max}$ or iAUC for capillary measurements, were observed.

An analysis of subgroups was performed in the PP population. Pronounced effects were observed in the middle-aged group (40–50 years old, n = 23) with a 30% reduction in venous glucose $iAUC_{0-120\ min}$ (p = 0.006) and a 5% reduction in $C_{max}$ (p = 0.004) after the supplement

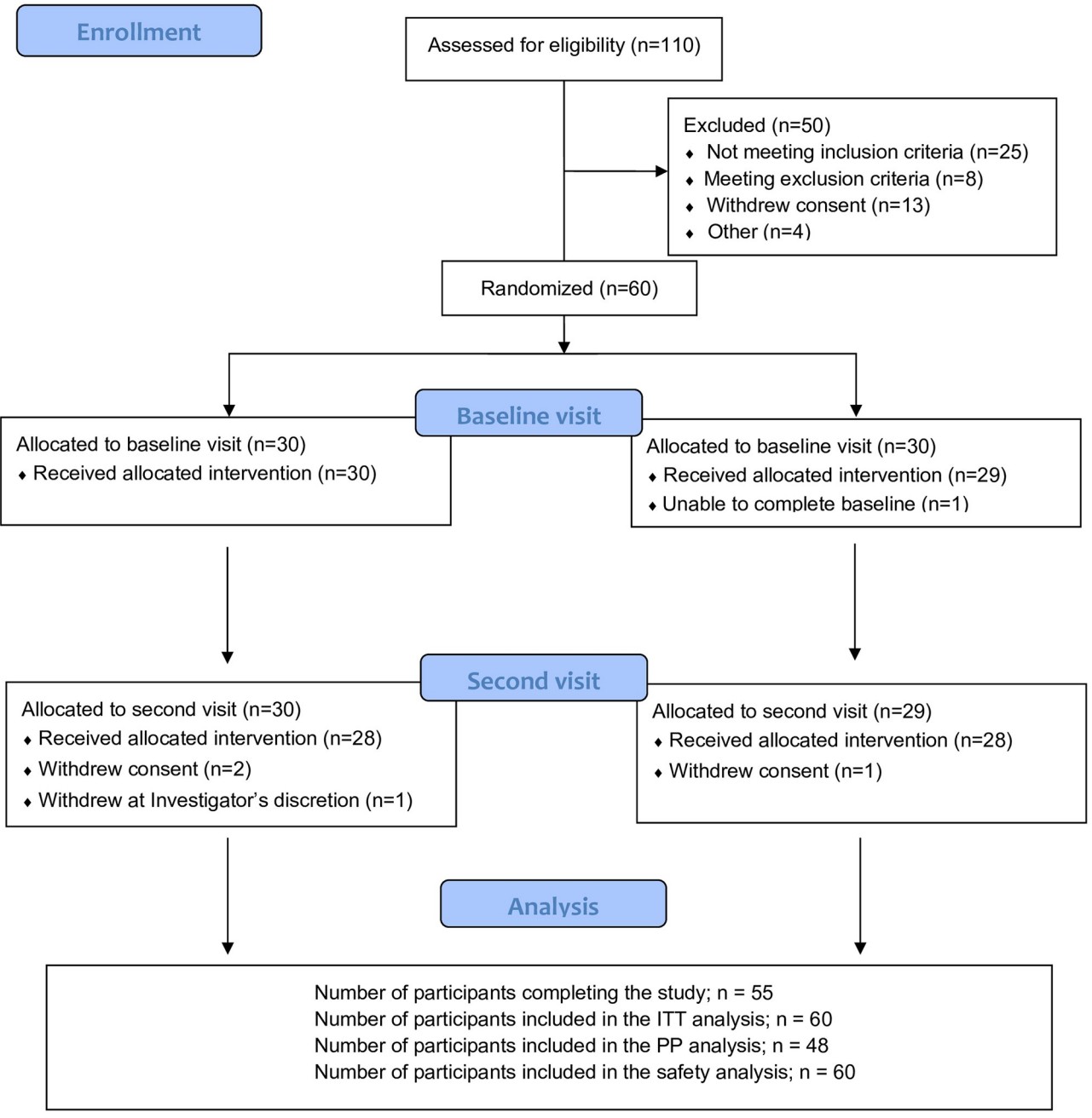

**Fig 1. Participant flow.**

compared to placebo. The other subgroups (gender, race) did not show any significant changes in the primary endpoint.

There were no severe adverse events, as categorized by the Medical Dictionary for Regulatory Activities, version 17. Out of 19 mild to moderate adverse events (AE) recorded in 16 individuals experiencing these events, three events were categorized as possibly related to the supplement (one nervous system disorder and two gastrointestinal disorders) and one as

**Table 1. Participant characteristics at screening[1].**

|  | ITT population (*n* = 60) | PP population (*n* = 48) |
|---|---|---|
| Gender, *n* (%) |  |  |
| Female | 32 (53%) | 23 (48%) |
| Male | 28 (47%) | 25 (52%) |
| Alcohol consumption status, *n* (%) |  |  |
| None | 21 (35%) | 17 (35%) |
| Occasional | 26 (43%) | 21 (44%) |
| Weekly | 13 (22%) | 10 (21%) |
| Smoking status, *n* (%) |  |  |
| Ex-smoker | 5 (8%) | 3 (6%) |
| No | 55 (92%) | 45 (94%) |
| Health questionnaire, *n* (%) |  |  |
| Excellent | 22 (37%) | 18 (38%) |
| Very good | 27 (45%) | 21 (44%) |
| Good | 11 (18%) | 9 (19%) |
| Race, *n* (%) |  |  |
| Western European white | 28 (47%) | 21 (44%) |
| Black or African American | 15 (25%) | 12 (25%) |
| Other | 17 (28%) | 15 (31%) |
| Age, yrs (SD) | 34.6 (10.4) | 35.5 (10.5) |
| Weight, kg (SD) | 79.7 (10.2) | 80.8 (10.7) |
| BMI, kg/m$^2$ (SD) | 27.4 (1.59) | 27.5 (1.59) |
| Fasting venous blood glucose, mmol/L (SD) | 4.93 (0.42) | 4.95 (0.42) |
| HbA1c, % (SD) | 5.48 (0.33) | 5.47 (0.31) |
| Systolic blood pressure, mmHg (SD) | 119.8 (8.9) | 117.7 (9.0) |
| Diastolic blood pressure, mmHg (SD) | 76.2 (8.1) | 75.8 (8.7) |
| Heart rate, bpm (SD) | 69.0 (11.1) | 69.2 (11.2) |

[1] Values are *n* (%), or means (SD)

possibly related to placebo (vascular disorder). All AEs were resolved by the end of the study without requiring medical treatment or hospitalization.

## Discussion

In this controlled clinical trial, we show a significant reduction of the serum glucose iAUC in healthy overweight subjects after a mixed meal consumed together with the supplement, compared to placebo. There are different ways in which the supplement could beneficially influence the glucose metabolism and health in the target population of individuals that are not yet affected by cardiovascular disease or diabetes. From a GI point of view [33], the supplement was able to reduce the overall GI of the high glycemic test meal. Although there is still some controversy on the role of low GI food and meals [12, 34], most studies comparing diets with different GI´s favor the ones with a lower GI in regards to the development of T2D [12–17, 35, 36].

Elevated postprandial glucose has been identified as an important risk factor for developing T2D as well as CVD and even overall mortality in individuals without impaired glucose tolerance (IGT) or T2D. The factual endpoints studied in these epidemiological studies were the 60

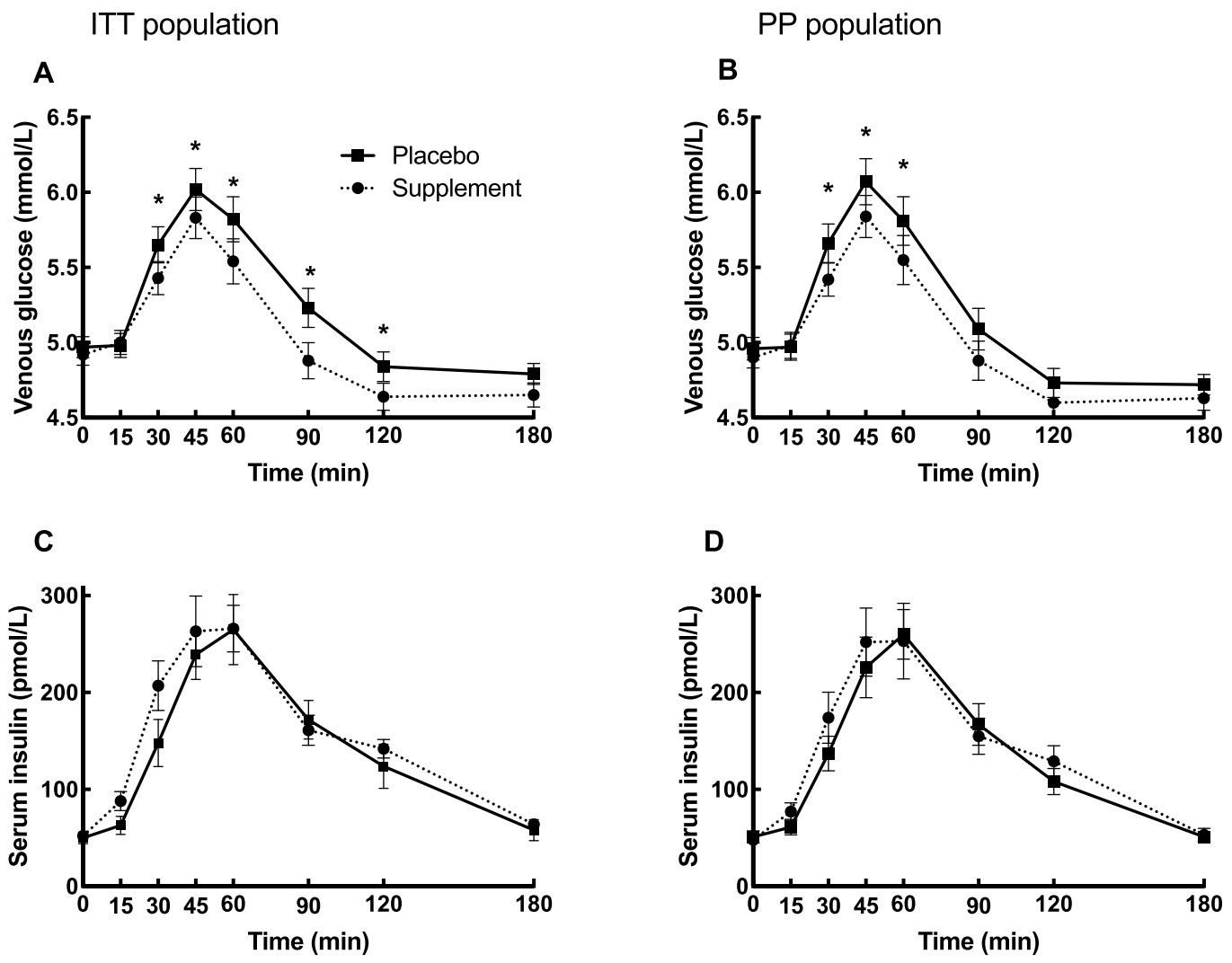

**Fig 2. Postprandial glucose and insulin responses.** Glucose and insulin responses for the ITT ($n = 54$–$58$, for different time points) and PP ($n = 48$) populations respectively after intake of a high carbohydrate sandwich meal with supplement or placebo. Data are expressed as means ± SE. Asterisc (*) indicates a significant difference between treatments ($p < 0.05$) at the respective time point.

and 120 minutes glucose values after a standardized glucose load varying between 50g and 100g of glucose [1–8]. This study showed significant differences for venous glucose at 30, 45, 60 and 90 minutes in favor of the supplement, thus showing a significant reduction of the surrogate risk factor 60-minute postprandial glucose. The use of a mixed meal as opposed to glucose solutions tests efficacy of the supplement closer to a real-life scenario.

There is ample evidence for a causal relationship between elevated postprandial glucose and the development of T2D and CVD. Experimental evidence on a cellular level shows that an induction of endothelial dysfunction by postprandial hyperglycemia is most likely a consequence of increased oxidative stress [18–21]. In human studies, postprandial glucose excursions have been shown to correlate with the extent of atherosclerosis [21]. Interventions targeting postprandial glycemia have been shown to prevent atherosclerosis but also the development of T2D [9–11] and possibly even cardiovascular events [9, 10].

**Table 2. Glucose and insulin responses for ITT and PP populations[1].**

| | Supplement | | | Placebo | | | Group Difference | | 95% CI | | |
|---|---|---|---|---|---|---|---|---|---|---|---|
| **ITT population** | n | Mean | SE | n | Mean | SE | Mean | SE | Upper | Lower | p-value |
| *Serum glucose* | | | | | | | | | | | |
| $iAUC_{0-120\ min}$, mmol/L•min | 54 | 52·20 | 6·79 | 58 | 68·72 | 6·62 | -16·54 | 6·71 | -3·08 | -30·01 | 0·017 |
| $iAUC_{0-180\ min}$, mmol/L•min | 54 | 62·44 | 8·42 | 58 | 79·07 | 8·19 | -16·63 | 8·56 | 0·55 | -33·81 | 0·058 |
| $C_{max0-120\ min}$, mmol/L | 54 | 6·10 | 0·12 | 58 | 6·45 | 0·12 | -0·35 | 0·09 | -0·17 | -0·53 | <0·001 |
| $T_{max0-120\ min}$, min | 54 | 51·04 | 2·49 | 58 | 52·29 | 2·85 | -1·25 | 5·32 | 5·32 | -7·82 | 0·704 |
| *Serum insulin* | | | | | | | | | | | |
| $iAUC_{0-120\ min}$, pmol/L•min | 54 | 15 803 | 1 589 | 58 | 14 523 | 1 556 | 1 279 | 1 334 | 3 958 | -1 399 | 0·342 |
| $iAUC_{0-180\ min}$, pmol/L•min | 54 | 19 361 | 1 908 | 58 | 17 253 | 1 869 | 2 108 | 1 589 | 5 298 | -1 083 | 0·191 |
| $C_{max\ 0-120\ min}$, pmol/L | 54 | 380·99 | 35·19 | 58 | 345·70 | 34·63 | 35·29 | 25·58 | 86·66 | -16·07 | 0·174 |
| $T_{max\ 0-120\ min}$, min | 54 | 59·38 | 3·43 | 58 | 58·78 | 3·33 | 0·60 | 3·76 | 8·16 | -6·95 | 0·873 |
| | **Supplement** | | | **Placebo** | | | **Group Difference** | | **CI 95%** | | |
| **PP population** | n | Mean | SE | n | Mean | SE | Mean | SE | Upper | Lower | p-value |
| *Serum glucose* | | | | | | | | | | | |
| $iAUC_{0-120\ min}$, mmol/L•min | 48 | 53·73 | 7·26 | 48 | 66·48 | 7·26 | -12·75 | 6·95 | 1·24 | -26·74 | 0·073 |
| $iAUC_{0-180\ min}$, mmol/L•min | 48 | 62·36 | 8·72 | 48 | 74·37 | 8·72 | -12·01 | 8·91 | 5·93 | -29·94 | 0·185 |
| $C_{max\ 0-120\ min}$, mmol/L | 48 | 6·12 | 0·14 | 48 | 6·42 | 0·14 | -0·29 | 0·09 | -0·12 | -0·47 | 0·002 |
| $T_{max\ 0-120\ min}$, min | 48 | 49·30 | 2·98 | 48 | 49·61 | 2·98 | -0·31 | 3·22 | 6·17 | -6·79 | 0·923 |
| *Serum insulin* | | | | | | | | | | | |
| $iAUC_{0-120\ min}$, pmol/L•min | 48 | 14 833 | 1 607 | 48 | 13 546 | 1 607 | 1 268 | 1 416 | 4 119 | -1 582 | 0·375 |
| $iAUC_{0-180\ min}$, pmol/L•min | 48 | 17 737 | 1 791 | 48 | 15 713 | 1 791 | 2 024 | 1 719 | 5 483 | -1 436 | 0·245 |
| $C_{max\ 0-120\ min}$, pmol/L | 48 | 357·15 | 35·81 | 48 | 324·42 | 35·81 | 32·73 | 26·62 | 86·30 | -20·85 | 0·225 |
| $T_{max\ 0-120\ min}$, min | 48 | 59·50 | 3·48 | 48 | 58·17 | 3·48 | 1·33 | 3·96 | 9·29 | -6·64 | 0·739 |

Participants in this study were mostly overweight and the glucose lowering effect of the supplement was pronounced in the older subgroup (40–50 yrs) showing a significant and impressive 30% reduction of $iAUC_{0-120\ min}$ glucose. Since overweight and increasing age are the main risk-factors for developing diabetes [1], efficacy of the supplement has been shown in a group, in which diabetes prevention is highly relevant.

The exact mechanisms leading to the observed effects of the supplement on postprandial glucose are not completely clear. Chromium is, however, known to play a role in glucose metabolism, possibly by potentiating insulin interaction with its receptor via binding of a low molecular chromium binding protein [37–39]. Yet, its role in improving glucose metabolism as a supplement is not well described in controlled clinical studies, most of which were performed in patients with T2D [40–43]. Milk and whey protein are known to stimulate insulin [44–50] and thus reduce the postprandial glucose response to a glucose solution [45, 47–51]. However, in the case of single amino acids, the insulinogenic effect is not attributable to all amino acids [52–55] and requires higher doses than the ones used here [52, 53, 56–58]. Hence, the iAUC for insulin was not significantly increased by the supplement compared to placebo and there was only an indication of a trend towards an earlier insulin response when looking at insulin iAUC´s. Studies looking at insulin and the development of T2D or IGT suggest that a pronounced first-phase insulin response compared to a late insulin increase could be beneficial [55, 59–61]. In addition it has been described, that proteins and amino acids have a distinct effect on gastric emptying, possibly delaying the glucose uptake [62–64]. A 'priming' of the stomach with the first sips of the supplement leading to a slowing of gastric emptying and a

better insulin homeostasis could contribute to the effects on postprandial glucose observed. Further research will have to substantiate these hypotheses.

The limitations of this study include missing values due to the high number of blood draws required for each participant and thus the need for imputation of some results when performing the iAUC calculation in the ITT analysis. Another limitation is the lack of data on menstrual cycle for the female participants, since it is known that blood glucose regulation differs between the follicular and luteal phases [65]. Overall, there were no significant effects when looking at the capillary blood measurements. This can be explained by the use of patient glucometers, which are known for their imprecision [66, 67] and cannot be expected to produce reliable data for the calculation of differences in glucose iAUC. This is surely a systematic problem for the analysis of this secondary end point. However, in a previous study of an early version of the supplement, professional glucometers used for capillary blood glucose measurements showed significant reductions of postprandial glucose, in line with that shown for venous glucose in this study [26]. Another limitation of this study was the variability in brands of bread used for the standardized test meal between the Canadian and American sites. This resulted in a slight difference in fiber content between the meals, with the Canadian brand having 2 grams more dietary fiber than the US brand. However, the macronutrient and caloric content of the test meals remained relatively similar between sites and individuals at each center received the same brand in the supplement and placebo test meal.

In conclusion, the supplement drink tested in this study efficiently reduced the postprandial glucose excursions after a meal with a high carbohydrate content without causing any adverse events. Future studies will have to evaluate whether regular consumption of the supplement improves markers of disease or might even prevent the development of diabetes. Effects on satiety and food cravings will also have to be studied. Definition of the mechanisms involved in the observed effects will help to develop supplements that could not only be used for the prevention of diabetes but may also be an option for dietary interventions in metabolic disease.

## Supporting information

**S1 Table. Inclusion and exclusion criteria.**
(PDF)

**S1 Checklist. Consort check-list.**
(DOC)

**S1 Data. Original data.**
(XLSX)

**S1 File.**
(PDF)

## Author Contributions

**Conceptualization:** Elin Östman, Azat Samigullin, Lovisa Heyman-Lindén, Kristina Andersson, Inger Björck, Rickard Öste, Per M. Humpert.

**Formal analysis:** Azat Samigullin, Lovisa Heyman-Lindén.

**Funding acquisition:** Rickard Öste.

**Methodology:** Elin Östman, Azat Samigullin, Lovisa Heyman-Lindén, Kristina Andersson, Inger Björck, Rickard Öste, Per M. Humpert.

**Resources:** Rickard Öste.

**Supervision:** Elin Östman, Inger Björck, Rickard Öste, Per M. Humpert.

**Visualization:** Elin Östman.

**Writing – original draft:** Elin Östman, Azat Samigullin, Per M. Humpert.

**Writing – review & editing:** Elin Östman, Azat Samigullin, Lovisa Heyman-Lindén, Kristina Andersson, Inger Björck, Rickard Öste, Per M. Humpert.

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
