## [Decision Letter · Decision Letter 0]

11 Nov 2019

PONE-D-19-15766

A novel nutritional supplement containing amino acids and chromium decreases postprandial glucose response in a randomized, double-blind, placebo-controlled study

PLOS ONE

Dear Dr Ostman,

Thank you for submitting your manuscript to PLOS ONE. After careful consideration, we feel that it has merit but does not fully meet PLOS ONE’s publication criteria as it currently stands. Therefore, we invite you to submit a revised version of the manuscript that addresses the points raised during the review process.

The reviewers raised several important points related to the design of the study, and suggestions for improving the level of detail and clarity in which you described methods of reporting the statistical analyses. Please carefully consider these points in your revision. There are some differences in opinion about the validity of some measures of glucose, and while I would appreciate you considering these, you may of course, disagree.

We would appreciate receiving your revised manuscript by Dec 23 2019 11:59PM. To enhance the reproducibility of your results, we recommend that if applicable you deposit your laboratory protocols in protocols.io, where a protocol can be assigned its own identifier (DOI) such that it can be cited independently in the future. For instructions see: http://journals.plos.org/plosone/s/submission-guidelines#loc-laboratory-protocols

We look forward to receiving your revised manuscript.

Kind regards,

Russell J de Souza, ScD, RD

Academic Editor

PLOS ONE

Journal Requirements:

2. Thank you for submitting your clinical trial to PLOS ONE and for providing the name of the registry and the registration number. The information in the registry entry suggests that your trial was registered after patient recruitment began. PLOS ONE strongly encourages authors to register all trials before recruiting the first participant in a study.

1) your reasons for your delay in registering this study (after enrolment of participants started);

2) confirmation that all related trials are registered by stating: “The authors confirm that all ongoing and related trials for this drug/intervention are registered”.

Please also ensure you report the date at which the ethics committee approved the study as well as the complete date range for patient recruitment and follow-up in the Methods section of your manuscript.

3. In your Methods section, please provide additional information about the participant recruitment method and on the health questionnaire used; moreover, please provide a link or a reference for the proprietary blend used.

4. Please provide the necessary documentation confirming that the IRB service used had jurisdiction in the United States of America.

5. Thank you for stating the following in the Financial Disclosure section:

'The study was funded in full by Aventure AB/Double Good AB, Sweden.'

We note that one or more of the authors are employed by a commercial company: starScience GmbH & Aventure AB.

Additional Editor Comments (if provided):

We note that in the demographic classification used, Asians, Hispanics and Aboriginal are included in the"North White Americans" group. As these populations show different risk levels for diabetes and CVDs, please clarify why it was chosen to group them together. 

Reviewers' comments:

Reviewer's Responses to Questions

**Comments to the Author**

1. Is the manuscript technically sound, and do the data support the conclusions?

Reviewer #1: Yes

Reviewer #2: Partly

Reviewer #3: Yes

2. Has the statistical analysis been performed appropriately and rigorously? 

Reviewer #1: Yes

Reviewer #2: Yes

Reviewer #3: Yes

3. Have the authors made all data underlying the findings in their manuscript fully available?

Reviewer #1: No

Reviewer #2: Yes

Reviewer #3: Yes

4. Is the manuscript presented in an intelligible fashion and written in standard English?

Reviewer #1: Yes

Reviewer #2: Yes

Reviewer #3: Yes

5. Review Comments to the Author

Reviewer #1: This is a well conducted and reported study. There were a number of issues related to the statistical design and analysis which I would recommend that the authors report differently, with reasons:

(1) The primary outcome should be explained a bit more clearly. The authors should, in a brief paragraph, describe how the iAUC is obtained. Where they cite methods of calculating it (e.g the trapezoid rule), they should explain why they have chosen this method instead of others.

(2) The study has several secondary outcomes. The authors should describe any considerations they have made to deal with multiple testing.

(3) The sample size calculation is poorly described. The authors say that it was calculated based on an expected effect size of 35.8mmol/L/min; however, this is not an effect size (a standardised difference). The authors should say what the expected mean value of the outcome was expected to be in the control group, along with the standard deviation and the difference in the mean outcome value expected due to the intervention, with justification.

(4) In the analysis of the outcome, the authors used a mixed model repeated measures ANCOVA with the pre-treatment value as a covariate. When using this approach, the effect of the pre-treatment value should be fixed to zero - the authors do not appear to have done this.

(5) In Table 1 the authors have presented a comparison of the characteristics of participants included in the ITT vs. PP populations. While this is useful information, what the authors should have presented in this table is a comparison of the same characteristics of individuals in the intervention group to those in the placebo group.

(6) In Table 2, the authors should report the mean value of the outcome and the standard errors (not standard deviation - SDs are reported for description (e.g. baseline tables), but SEs are reported for inference). They should also report the difference in outcome and the confidence intervals for the difference along with the p-value. Please see the report of other clinical trials, e.g. (https://journals.plos.org/plosmedicine/article/figure?id=10.1371/journal.pmed.1001518.t003) to see how this should be done.

(7) Instead of cite the treatment effect as a percent reduction in serum glucose, the authors should report the absolute reduction in serum glucose, along with the 95% confidence intervals and p-value. If the authors want to express this as a relative reduction, this should be done in the discussion, or as an alternative (but not replacement for the absolute reductions). If reporting the relative reductions, the confidence intervals for these reductions should also be reported.

(8) The authors should avoid the +/- designation when reporting numerical results as it is very misleading and actually quite meaningless in many of the contexts it has been used. For example, in the abstract, it is difficult to tell whether a decrease in glucose iAUC-120min 49.5 +/- 47 refers to decreases varying from 2.5 to 96.5 or whether 47 is the IQR. Similarly in Table 1, mean age in the ITT population is reported as 34.6 +/- 10.4 with a footnote indicating that the +/- are SD; however, what does a mean +/- the SD really mean in practical terms? This should not be reported in this manner. Instead, the abstract should be reporting the mean reductions and their 95% confidence intervals; this has a very clear statistical meaning. When reporting the table of characteristics where it is typical to report means and their SDs, the authors should simply report the mean followed by the SD in brackets and indicate so in the footnote, e.g. 34.6 (10.4). It may be common to use the +/- designation in the literature, but because of the inconsistent way in which this is done (some use it to mean ranges, others IQRs, yet others SDs, mostly without clear indication), it is best to avoid this. See https://journals.plos.org/plosmedicine/article/figure?id=10.1371/journal.pmed.1001594.t001 for an example of how to report means SDs and ranges for descriptive purposes.

Reviewer #2: Aim of the study was that to evaluate postprandial responses in serum glucose and insulin levels in a cohort of overweight subjects of both genders, treated with a supplement consisting in 2.6 g of amino acids (L-Leucine, L-Threonine, L-Lysine, L-Isoleucine and L-Valine) and 250 mcg of chromium picolinate vs. placebo. In the intention-to-treat-analysis (n = 60), the supplement significantly decreased glucose levels (as iAUC0-120 min, iAUC0-180 min and Cmax), without affecting those of insulin. The treatment was well tolerated.

The study is well designed and the results are adequately presented. Due to the paucity of biochemical parameters (serum glucose and insulin), the manuscript should be submitted as a rapid communication in a journal of nutrition sciences. Furthermore, some major and minor issues should be solved.

Comments

1. In the Introduction the rationale has not been clearly reported. In particular, there is no information regarding the reasons for which the Authors chose the specific ingredients of the supplement. For instance, the gastroenteropharmacological properties of the selected amino acids (branched-chain amino acids?) should be mentioned.

2. Few biochemical parameters have been measured. Only serum levels of glucose and insulin. To understand the biochemical or pharmacological mechanisms underlying the hypoglycemic effect of the supplement, the Authors should evaluate circulating levels of some gastrointestinal peptides, particularly incretins such as GLP-1, which exert insulinotropic effects and reduce gastric empting. See the article by Rigamonti et al., Nutrients, 2019;11(2). pii: E247. doi: 10.3390/nu11020247.

3. The results referring to glycaemia measured with glucometer, which is defined an “imprecise” method by the Authors (pag. 17 – line 81), should be removed, because they do not add anything.

4. The Authors should explain the differences in results obtained in the per-protocol analysis vs in the intention-to-treat analysis.

5. In the Abstract the Authors should state that subjects were overweight and of both genders. Not only in the conclusions.

6. The Authors should report the phase of the menstrual cycle during which the experimental tests were performed in female subjects.

7. In the final part of the Discussion, when reporting limitations of the study, the Authors state that fiber content in bread used in the two centers is different. An appropriate statistical analysis should rule out the potential “interfering” effect of this factor.

8. Numbers of females/males should be reported in the section of Participants (pag. 4).

9. The exact doses of the single amino acids should be indicated. Not simply 2.6 g of amino acids (L-Leucine, L-Threonine, L-Lysine, L-Isoleucine and L-Valine).

10. Pag. 5: protein/fat should be replaced with proteins/fats.

11. Sensitivity and intra-/inter-assay coefficients of variation of the analytical methods for glucose and insulin should be indicated.

12. Pag. 6; the safety endpoints were evaluated at the three visits. Please, specify these time points.

13. Fig. 1 is of low resolution and difficult to read. Please, improve the quality of this figure.

14. Pag. 14 – Line 1: “Lead” should be “led”.

15. Pag. 14 – Line 16; what are the “other subgroups”? Please, specify.

16. Pag. 14 – Line 19: the Authors state that adverse events were categorized as possibly related to supplement or placebo. What system of categorization (or algorithm) was used?

17. Pag. 15 – line 33: where was the acronym IGT (impaired glucose tolerance) abbreviated?

18. Pag. 16 – lines 65-68: the Authors discuss the results by citing “unpublished” results. This is scientifically not appropriate.

Reviewer #3: This clinical trial was designed to evaluate the efficacy of a nutritional supplement to lower blood glucose levels after a meal in health individuals. The study was well-conducted and the data were clearly described. I only have few comments, but that should be included in the revised manuscript:

1) A conflicting result is regarding the lack of changes when blood glucose levels were evaluated using glucometers, compared to enzymatic methods that determined serum glucose. The authors argued that regular glucometers used in the present study are imprecise, but I personally think that this argument is not necessarily valid. Anyway, I think that the authors should include the information of the lack of differences in blood glucose levels when using glucometers in the Abstract. Thus, the readers can be informed that the improvement in glucose homeostasis was verified by one method of analysis but was not confirmed by data obtained using glucometers.

2) The adverse events should be described in Results. They are only briefly mentioned without explaining what they refer to.

3) Although I agree that this is not required for this clinical trial, I recommend that the authors measure GI-secreted hormones (e.g., GLP-1, PYY, GIP or ghrelin) during their trials to try to determine whether the supplement could have its effects via humoral mechanisms.

6. PLOS authors have the option to publish the peer review history of their article (what does this mean?). If published, this will include your full peer review and any attached files.

Reviewer #1: No

Reviewer #2: No

Reviewer #3: Yes: Jose Donato Junior

---

## [Author Response · Author response to Decision Letter 0]

13 Jan 2020

Journal Requirements

Answer: All format requirements have been reviewed and manuscript revised accordingly.

2. Thank you for submitting your clinical trial to PLOS ONE and for providing the name of the registry and the registration number. The information in the registry entry suggests that your trial was registered after patient recruitment began. PLOS ONE strongly encourages authors to register all trials before recruiting the first participant in a study.

1) your reasons for your delay in registering this study (after enrolment of participants started);

2) confirmation that all related trials are registered by stating: “The authors confirm that all ongoing and related trials for this drug/intervention are registered”.

Please also ensure you report the date at which the ethics committee approved the study as well as the complete date range for patient recruitment and follow-up in the Methods section of your manuscript.

Answer: The task of registering the study had unfortunately fallen between chairs of the authors and the contracted research organization and that caused the delay. A sentence explaining this have been added on page 4. The date of IRB approval is stated on page 4. The date range for recruitment and follow-up is stated under Results on page 9.

3. In your Methods section, please provide additional information about the participant recruitment method and on the health questionnaire used; moreover, please provide a link or a reference for the proprietary blend used. 

Answer: Information on recruitment method and self rated health has been added on pages 4-5. Reference to previous publication of proprietary blend has been added on page 5. 

4. Please provide the necessary documentation confirming that the IRB service used had jurisdiction in the United States of America. 

Answer: In the uploaded IRB approval of 16 March 2017 there is a section called Compliance statement/Attestation where it is stated that the IRBs of IRB Services are registered with OHRP and FDA. According to the CRO used for the study, IRB Services Inc (now Advarra) was a part of Cheasepeake Bay IRB (a US company). It has jurisdiction in both countries.

5. Thank you for stating the following in the Financial Disclosure section:

'The study was funded in full by Aventure AB/Double Good AB, Sweden.'

We note that one or more of the authors are employed by a commercial company: starScience GmbH & Aventure AB.

Within your Competing Interests Statement, please confirm that this commercial affiliation does not alter your adherence to all PLOS ONE policies on sharing data and materials by including the following statement: "This does not alter our adherence to PLOS ONE policies on sharing data and materials.” (as detailed online in our guide for authorshttp://journals.plos.org/plosone/s/competing-interests) . If this adherence statement is not accurate and there are restrictions on sharing of data and/or materials, please state these. Please note that we cannot proceed with consideration of your article until this information has been declared.

Answer: Funding and Competing Interests Statements have been updated in the cover letter.

Additional Editor Comments (if provided):

We note that in the demographic classification used, Asians, Hispanics and Aboriginal are included in the"North White Americans" group. As these populations show different risk levels for diabetes and CVDs, please clarify why it was chosen to group them together. 

Answer: The aim of the study was to recruit 50% “North White Americans” and 50% “African Americans” and not making any further classifications. It turned out to be more difficult then anticipated to recruit African Americans and despite numerous efforts we ended up with only 25% African Americans, making that group too small to draw firm conclusions from. The “North White Americans” are presented as 47% “Western European Whites” and 28% “Other” but no classifications were used for the primary analysis in the study. It would not make sense from a statistical power point of view.

Response to reviewers

Reviewer #1: This is a well conducted and reported study. There were a number of issues related to the statistical design and analysis which I would recommend that the authors report differently, with reasons:

(1) The primary outcome should be explained a bit more clearly. The authors should, in a brief paragraph, describe how the iAUC is obtained. Where they cite methods of calculating it (e.g the trapezoid rule), they should explain why they have chosen this method instead of others.

Answer: The trapezoid rule was used for calculating iAUC0-120min and iAUC0-180min. The trapezoidal rule is a numerical integration method used to approximate the area under a curve. 

Explanations and references have been added on pages 6 and 8.

(2) The study has several secondary outcomes. The authors should describe any considerations they have made to deal with multiple testing.

Answer: The secondary outcomes are all aimed at different aspects of postprandial glycemia as it is the primary endpoint. They can be clustered in venous results, capillary results and insulin. The magnitude and significance of the secondary endpoints in this case can either support or oppose the result obtained as the primary endpoint. We present the results and discuss these accordingly. 

(3) The sample size calculation is poorly described. The authors say that it was calculated based on an expected effect size of 35.8mmol/L/min; however, this is not an effect size (a standardized difference). The authors should say what the expected mean value of the outcome was expected to be in the control group, along with the standard deviation and the difference in the mean outcome value expected due to the intervention, with justification.

Answer: The description has been revised.

(4) In the analysis of the outcome, the authors used a mixed model repeated measures ANCOVA with the pre-treatment value as a covariate. When using this approach, the effect of the pre-treatment value should be fixed to zero - the authors do not appear to have done this.

Answer: We have contacted the responsible CRO (KGK Science) and addressed the issue with the responsible biostatistician. The biostatistician informed us that there was an error in the methods description in the study report and that in fact an ANOVA and not an ANCOVA was performed (thus no covariate was involved in the first place). In this process the authors asked the CRO to provide additional details on the iAUC calculation and noted that in the ITT group iAUC has been calculated in 3 cases despite missing adjacent glucose values. These 3 iAUC values were excluded from the analysis and the biostatistician provided updated results (see Table 2, Figure 2). Some strength of the association was lost in the process, but statistical significance remained. The original data including iAUCs calculated by the CRO will be provided.

The method used has been updated/revised accordingly.

(5) In Table 1 the authors have presented a comparison of the characteristics of participants included in the ITT vs. PP populations. While this is useful information, what the authors should have presented in this table is a comparison of the same characteristics of individuals in the intervention group to those in the placebo group.

Answer: Since it is a cross-over study, all participants had both the supplement and placebo meals. Thus, there are no comparisons to be made between groups and we find it informative to show how the characteristics change from the higher number of participants included in the ITT population compared with the reduced number of participants in the PP population.

(6) In Table 2, the authors should report the mean value of the outcome and the standard errors (not standard deviation - SDs are reported for description (e.g. baseline tables), but SEs are reported for inference). They should also report the difference in outcome and the confidence intervals for the difference along with the p-value. Please see the report of other clinical trials, e.g. (https://journals.plos.org/plosmedicine/article/figure?id=10.1371/journal.pmed.1001518.t003) to see how this should be done.

Answer: Table 2 has been revised accordingly.

(7) Instead of citing the treatment effect as a percent reduction in serum glucose, the authors should report the absolute reduction in serum glucose, along with the 95% confidence intervals and p-value. If the authors want to express this as a relative reduction, this should be done in the discussion, or as an alternative (but not replacement for the absolute reductions). If reporting the relative reductions, the confidence intervals for these reductions should also be reported.

Answer: Presentation of results have been revised accordingly. 

(8) The authors should avoid the +/- designation when reporting numerical results as it is very misleading and actually quite meaningless in many of the contexts it has been used. For example, in the abstract, it is difficult to tell whether a decrease in glucose iAUC-120min 49.5 +/- 47 refers to decreases varying from 2.5 to 96.5 or whether 47 is the IQR. Similarily in Table 1, mean age in the ITT population is reported as 34.6 +/- 10.4 with a footnote indicating that the +/- are SD; however, what does a mean +/- the SD really mean in practical terms? This should not be reported in this manner. Instead, the abstract should be reporting the mean reductions and their 95% confidence intervals; this has a very clear statistical meaning. When reporting the table of characteristics where it is typical to report means and their SDs, the authors should simply report the mean followed by the SD in brackets and indicate so in the footnote, e.g. 34.6 (10.4). It may be common to use the +/- designation in the literature, but because of the inconsistent way in which this is done (some use it to mean ranges, others IQRs, yet others SDs, mostly without clear indication), it is best to avoid this. See https://journals.plos.org/plosmedicine/article/figure?id=10.1371/journal.pmed.1001594.t001 for an example of how to report means SDs and ranges for descriptive purposes.

Answer: Table 1 as well as the Abstract has been adjusted according to the suggestion from the reviewer. Since the values were not normally distributed the statistician decided that a rank transformation prior to ANOVA was necessary. The ANOVA has thus been run on rank-transformed variables and in our opinion, 95% confidence intervals of the ranks would not be helpful to the reader.

 

Reviewer #2: Aim of the study was that to evaluate postprandial responses in serum glucose and insulin levels in a cohort of overweight subjects of both genders, treated with a supplement consisting in 2.6 g of amino acids (L-Leucine, L-Threonine, L-Lysine, L-Isoleucine and L-Valine) and 250 mcg of chromium picolinate vs. placebo. In the intention-to-treat-analysis (n = 60), the supplement significantly decreased glucose levels (as iAUC0-120 min, iAUC0-180 min and Cmax), without affecting those of insulin. The treatment was well tolerated.

The study is well designed, and the results are adequately presented. Due to the paucity of biochemical parameters (serum glucose and insulin), the manuscript should be submitted as a rapid communication in a journal of nutrition sciences. Furthermore, some major and minor issues should be solved.

Comments

1. In the Introduction the rationale has not been clearly reported. In particular, there is no information regarding the reasons for which the Authors chose the specific ingredients of the supplement. For instance, the gastroenteropharmacological properties of the selected amino acids (branched-chain amino acids?) should be mentioned.

Answer: One sentence has been added to the introduction to reference previous work done with the respective ingredients. Furthermore, it is mentioned in the discussion (lines 66-67) that “proteins and amino acids have a distinct effect on gastric emptying, possibly delaying the glucose uptake”.

2. Few biochemical parameters have been measured. Only serum levels of glucose and insulin. To understand the biochemical or pharmacological mechanisms underlying the hypoglycemic effect of the supplement, the Authors should evaluate circulating levels of some gastrointestinal peptides, particularly incretins such as GLP-1, which exert insulinotropic effects and reduce gastric empting. See the article by Rigamonti et al., Nutrients, 2019;11(2). pii: E247. doi: 10.3390/nu11020247.

Answer: No adequate samples are available and thus no further analysis can be made to study effects on gut hormones. However, we are planning to analyze GLP-1 and glucagon in upcoming studies.

3. The results referring to glycaemia measured with glucometer, which is defined an “imprecise” method by the Authors (pag. 17 – line 81), should be removed, because they do not add anything.

Answer: Reviewer 3 has an opposing view on this. We have kept the information in the manuscript and added clarification in the Abstract (see comments below).

4. The Authors should explain the differences in results obtained in the per-protocol analysis vs in the intention-to-treat analysis.

Answer: There are hardly any differences in characteristics between the ITT and PP groups. Thus, the likely explanation for differences in results is assumed to be related to the reduced number of subjects in the PP population.

5. In the Abstract the Authors should state that subjects were overweight and of both genders. Not only in the conclusions.

Answer: We already stated the number of females and males in the abstract and have added the fact that they were overweight.

6. The Authors should report the phase of the menstrual cycle during which the experimental tests were performed in female subjects. 

Answer: The only data available on menstrual cycle, is the “last date of their menstrual cycle”. Unfortunately, no additional data on menstrual cycle was collected. A literature reference and description has been added to the Limitations paragraph in the Discussion.

7. In the final part of the Discussion, when reporting limitations of the study, the Authors state that fiber content in bread used in the two centers is different. An appropriate statistical analysis should rule out the potential “interfering” effect of this factor.

Answer: We don´t see a point in correcting for fiber differences, since each participant had the same fiber content in both test meals as a consequence of the cross–over design.

8. Numbers of females/males should be reported in the section of Participants (pag. 4).

Answer: Since the section about Participants on page 4 mainly describe the inclusion and exclusion criteria, we think the better place to specify the number of female and male participants is under the Results section on page 9, so we did.

9. The exact doses of the single amino acids should be indicated. Not simply 2.6 g of amino acids (L-Leucine, L-Threonine, L-Lysine, L-Isoleucine and L-Valine).

Answer: This information is of sensitive nature and protected by patent. Details have thus not been included in the manuscript. We are in contact with the editorial office to clarify this point. 

10. Pag. 5: protein/fat should be replaced with proteins/fats.

Answer: Has been changed accordingly.

11. Sensitivity and intra-/inter-assay coefficients of variation of the analytical methods for glucose and insulin should be indicated.

Answer: Information has been added.

12. Pag. 6; the safety endpoints were evaluated at the three visits. Please, specify these time points.

Answer: Clarification has been added.

13. Fig. 1 is of low resolution and difficult to read. Please, improve the quality of this figure.

Answer: The resolution has been increased.

14. Pag. 14 – Line 1: “Lead” should be “led”. 

Answer: has been changed accordingly.

15. Pag. 14 – Line 16; what are the “other subgroups”? Please, specify. 

Answer: has been changed accordingly.

16. Pag. 14 – Line 19: The Authors state that adverse events were categorized as possibly related to supplement or placebo. What system of categorization (or algorithm) was used?

Answer: Clarification has been added. 

17. Pag. 15 – line 33: where was the acronym IGT (impaired glucose tolerance) abbreviated? 

Answer: The acronym has been spelled out.

18. Pag. 16 – lines 65-68: The Authors discuss the results by citing “unpublished” results. This is scientifically not appropriate.

Answer: The section has been removed. 

Reviewer #3: This clinical trial was designed to evaluate the efficacy of a nutritional supplement to lower blood glucose levels after a meal in health individuals. The study was well-conducted, and the data were clearly described. I only have few comments, but that should be included in the revised manuscript:

1) A conflicting result is regarding the lack of changes when blood glucose levels were evaluated using glucometers, compared to enzymatic methods that determined serum glucose. The authors argued that regular glucometers used in the present study are imprecise, but I personally think that this argument is not necessarily valid. Anyway, I think that the authors should include the information of the lack of differences in blood glucose levels when using glucometers in the Abstract. Thus, the readers can be informed that the improvement in glucose homeostasis was verified by one method of analysis but was not confirmed by data obtained using glucometers.

Answer: We have added this information accordingly

2) The adverse events should be described in Results. They are only briefly mentioned without explaining what they refer to.

Answer: This information has been added.

3) Although I agree that this is not required for this clinical trial, I recommend that the authors measure GI-secreted hormones (e.g., GLP-1, PYY, GIP or ghrelin) during their trials to try to determine whether the supplement could have its effects via humoral mechanisms.

Answer: No adequate samples are available and thus no further analysis can be made to study effects on gut hormones. However, we are planning to analyze GLP-1 and glucagon in upcoming studies.

---

## [Decision Letter · Decision Letter 1]

13 Mar 2020

PONE-D-19-15766R1

A novel nutritional supplement containing amino acids and chromium decreases postprandial glucose response in a randomized, double-blind, placebo-controlled study

PLOS ONE

Dear Dr Ostman,

Thank you for submitting your manuscript to PLOS ONE. After careful consideration, we feel that it has merit but does not fully meet PLOS ONE’s publication criteria as it currently stands. Therefore, we invite you to submit a revised version of the manuscript that addresses the points raised during the review process.

Please respond to the Reviewer's comments including explanation regarding patent-protected information. 

We would appreciate receiving your revised manuscript by Apr 27 2020 11:59PM. To enhance the reproducibility of your results, we recommend that if applicable you deposit your laboratory protocols in protocols.io, where a protocol can be assigned its own identifier (DOI) such that it can be cited independently in the future. For instructions see: http://journals.plos.org/plosone/s/submission-guidelines#loc-laboratory-protocols

We look forward to receiving your revised manuscript.

Kind regards,

Maciej S. Buchowski

Academic Editor

PLOS ONE

Reviewers' comments:

Reviewer's Responses to Questions

**Comments to the Author**

1. If the authors have adequately addressed your comments raised in a previous round of review and you feel that this manuscript is now acceptable for publication, you may indicate that here to bypass the “Comments to the Author” section, enter your conflict of interest statement in the “Confidential to Editor” section, and submit your "Accept" recommendation.

Reviewer #1: (No Response)

Reviewer #2: (No Response)

Reviewer #3: All comments have been addressed

2. Is the manuscript technically sound, and do the data support the conclusions?

Reviewer #1: Yes

Reviewer #2: Yes

Reviewer #3: Yes

3. Has the statistical analysis been performed appropriately and rigorously? 

Reviewer #1: Yes

Reviewer #2: Yes

Reviewer #3: Yes

4. Have the authors made all data underlying the findings in their manuscript fully available?

Reviewer #1: Yes

Reviewer #2: No

Reviewer #3: Yes

5. Is the manuscript presented in an intelligible fashion and written in standard English?

Reviewer #1: Yes

Reviewer #2: Yes

Reviewer #3: Yes

6. Review Comments to the Author

Reviewer #1: This manuscript is vastly improved since the previous round of peer review, and I commend the authors for taking on board most of the reviewers' suggestions. There is still one major issue which the authors have not addressed, and a few minor ones which have arisen since the last round.

Major issues:

- in my comment (6) in the previous round, I made a recommendation on how authors should report means and SDs in the descriptive tables, and group means and SEs in the inference tables, along with effects as between-group differences with their 95% confidence intervals and p-values. The authors have done most of this, but have not reported the 95% confidence intervals for the differences/effects; this is very important and should be done. Even where authors have decided to report effects in relative terms (e.g. 24.7% reduction in venous serum iAUC), a 95% confidence interval for this effect should be reported - although I would generally recommend that you stick to absolute differences in the results and perhaps keep relative differences to the discussion, because your analytical method does not estimate relative differences. Reporting SEs and SDs is not necessary in the abstract, but the 95% confidence intervals are crucial.

- the authors report some rank transformation technique to calculate p-values due to non-normality of 'values' (I'm assuming you mean outcome measures here). Normality of outcomes is not a requisite to report p-values from an ANOVA/regression; it is only the normality of residuals that is assumed.

Other points:

Abstract

- If you choose to report the SEs in the abstract despite the comment above, please indicate what quantities are in the brackets (in addition to specifying the confidence intervals). Also, consider rephrasing all through to refer to 'differences' rather than 'reductions'/'decreases' in quantities. For example, you could say: "... the supplement resulted in decreased venous blood glucose compared to placebo, mean (SE) of 69.3 (7.2) versus 52.2 (6.4) respectively, a difference of 17.1 mmol/L (95%CI X to Y, p-value x.xxx)". Please report the exact p-values unless they are less than 0.001.

- the following phrase is completely inappropriate and should be removed: "... missed statistical significance". It implies that statistical significance is some magical target to be hit or missed which is not the case. You either find evidence of a difference, based on a statistically significant finding, or you don't find that evidence; but phraseology such as this suggesting that you aimed to find a difference but missed it is very poor statistical reporting.

Methods

- p5 line 94: do you mean "... a total of 2.6g of five amino acids (L-leucine..."

- p6, 'Outcome measures': the iAUC-0,120min could still be better described than has been done so far. Does it represent a change in plasma glucose concentration up to 120 minutes after ingestion of the meal/product? If so, describe it as such (or similar terms). Same for the other iAUCs.

- p8 second paragraph of 'statistical analysis' - there appears to be an incomplete sentence beginning "Although simple, the trapezoidal rule..."

- p8 line 178: you mean "complete observations on 57 individuals", not complete datasets.

- p8 line 180: the paragraph on page 9 from line 184 should be moved to the sentence that ends with "imputation." on line 180. After that, the next sentence should begin with p-values, not probabilities, which has a very different meaning in this context.

- p9 line 187 - see comment above about normality.

- Table 2 - see comment above about reporting differences with their 95% confidence intervals and p-values

- p14 line 8 - see comment above about "missing statistical significance", which is a very serious and important point about statistical reporting.

Reviewer #2: The Authors have minimally responded to the Referee’s comments.

As no new data have been added in the revised version, the Referee again suggests the Authors to submit the manuscript as a rapid communication to a journal dealing with nutrition and related topics.

The Referee does not hide the disappoint due to the Authors’ refusal to report the amounts of amino acids present in the supplement, which the Authors declare to be protected by a patent.

Reviewer #3: (No Response)

7. PLOS authors have the option to publish the peer review history of their article (what does this mean?). If published, this will include your full peer review and any attached files.

Reviewer #1: No

Reviewer #2: No

Reviewer #3: No

---

## [Author Response · Author response to Decision Letter 1]

24 Apr 2020

Reviewer #1: 

This manuscript is vastly improved since the previous round of peer review, and I commend the authors for taking on board most of the reviewers' suggestions. There is still one major issue which the authors have not addressed, and a few minor ones which have arisen since the last round.

Major issues:

- in my comment (6) in the previous round, I made a recommendation on how authors should report means and SDs in the descriptive tables, and group means and SEs in the inference tables, along with effects as between-group differences with their 95% confidence intervals and p-values. The authors have done most of this, but have not reported the 95% confidence intervals for the differences/effects; this is very important and should be done. Even where authors have decided to report effects in relative terms (e.g. 24.7% reduction in venous serum iAUC), a 95% confidence interval for this effect should be reported - although I would generally recommend that you stick to absolute differences in the results and perhaps keep relative differences to the discussion, because your analytical method does not estimate relative differences. Reporting SEs and SDs is not necessary in the abstract, but the 95% confidence intervals are crucial.

- the authors report some rank transformation technique to calculate p-values due to non-normality of 'values' (I'm assuming you mean outcome measures here). Normality of outcomes is not a requisite to report p-values from an ANOVA/regression; it is only the normality of residuals that is assumed.

Answer:

We apologize for the imprecision of our last response. The rank transformation had been performed due to non-normality of both the data and the residuals. Ranking data, as was performed in the analysis, is one of the options for handling this issue (1,2). But our statistician advised that presenting the 95% confidence intervals of the non-ranked transformed data with ranked p-values has limited statistical value. 

However, we have found that some literature explicitly recommends against examining data and residuals for normality (3), especially since the F-test for sample sizes larger than 20 or 30 where there are equal sample sizes per study arm (which we have) is robust against moderate departures from the assumptions of normality and homogeneity of variances (4). Thus, running the same test without prior ranking is also a plausible option, which would ensure consistency with the confidence interval (i.e. linear mixed model with treatment as main effect, subject number as random effect controlling sequence and period instead of a ranked linear mixed model with treatment as main effect, subject number as random effect controlling sequence and period). As could be expected, the p-values and means derived from this analysis are slightly different from the ranked data, since they are least squared means that have been adjusted for subject random effect, sequence and period. Hence, we have adapted the manuscript accordingly in the text and Table 2.

(1) Conover W. J. in Lovric M. International Encyclopedia of Statistical Science: Springer 2011, p. 1184-1193

(2) Sawilowsky S. S. Nonparametric tests of interaction in experimental design. Review of Educational Research. 1990; 60: 91-126

(3) Gelman A, Hill J. Data analysis using regression and multilevel/hierarchical models: Cambridge university press 2006. P 45-46

(4) Cohen J. Statistical power analysis for the behavioral sciences. 2 edn: Lawrence Erlbaum Associates, Publishers 1988, p. 407-408

Other points:

Abstract

- If you choose to report the SEs in the abstract despite the comment above, please indicate what quantities are in the brackets (in addition to specifying the confidence intervals). Also, consider rephrasing all through to refer to 'differences' rather than 'reductions'/'decreases' in quantities. For example, you could say: "... the supplement resulted in decreased venous blood glucose compared to placebo, mean (SE) of 69.3 (7.2) versus 52.2 (6.4) respectively, a difference of 17.1 mmol/L (95%CI X to Y, p-value x.xxx)". Please report the exact p-values unless they are less than 0.001.

Answer:

The phrasing in the abstract has been revised according to the suggestions.

- the following phrase is completely inappropriate and should be removed: "... missed statistical significance". It implies that statistical significance is some magical target to be hit or missed which is not the case. You either find evidence of a difference, based on a statistically significant finding, or you don't find that evidence; but phraseology such as this suggesting that you aimed to find a difference but missed it is very poor statistical reporting.

Answer: 

The phrase has been removed.

Methods

- p5 line 94: do you mean "... a total of 2.6g of five amino acids (L-leucine...

Answer: 

We specify the meaning of 5AA in the introduction and thus think it is appropriate to use that abbreviation when we talk about the exact same blend. Consequently, we kept the text unchanged.

- p6, 'Outcome measures': the iAUC-0,120min could still be better described than has been done so far. Does it represent a change in plasma glucose concentration up to 120 minutes after ingestion of the meal/product? If so, describe it as such (or similar terms). Same for the other iAUCs.

Answer: 

An extra sentence has been added for clarification.

- p8 second paragraph of 'statistical analysis' - there appears to be an incomplete sentence beginning "Although simple, the trapezoidal rule..."

Answer: 

We cannot see that the sentence is incomplete but have tried to rephrase to make it more clear. 

- p8 line 178: you mean "complete observations on 57 individuals", not complete datasets.

Answer: 

The sentence has been rephrased to clarify.

- p8 line 180: the paragraph on page 9 from line 184 should be moved to the sentence that ends with "imputation." on line 180. After that, the next sentence should begin with p-values, not probabilities, which has a very different meaning in this context.

Answer: 

The paragraphs were restructured according to the comment.

- p9 line 187 - see comment above about normality.

Answer:

Sentence was removed, since statistical method was revised according to Reviewers suggestion. 

- Table 2 - see comment above about reporting differences with their 95% confidence intervals and p-values

Answer:

95% confidence intervals were included.

- p14 line 8 - see comment above about "missing statistical significance", which is a very serious and important point about statistical reporting.

Answer:

The sentences have been rephrased.

Reviewer #2: 

The Authors have minimally responded to the Referee’s comments.

As no new data have been added in the revised version, the Referee again suggests the Authors to submit the manuscript as a rapid communication to a journal dealing with nutrition and related topics.

Answer:

The manuscript has been under Review for publication in PLOS one for a very long time after recommendation by the board of PLOS Med. We sincerely hope to qualify for publication now, a novel submission to a different journal would definitely be a non-preferred option. In addition, we believe that PLOSone is the right journal for this study since it allows wide availability to the scientific community in a research area that affects millions of individuals worldwide.

The Referee does not hide the disappoint due to the Authors’ refusal to report the amounts of amino acids present in the supplement, which the Authors declare to be protected by a patent.

Answer:

We understand the Reviewers disappointment in this matter; however, we hope that there is some understanding that scientific work and investment of many years needs to be protected from patent infringement. We are very happy to share material or information after publication of the manuscript with individual scientists/researchers upon request. 

We thank the Reviewer for working on this manuscript.

---

## [Decision Letter · Decision Letter 2]

22 May 2020

A novel nutritional supplement containing amino acids and chromium decreases postprandial glucose response in a randomized, double-blind, placebo-controlled study

PONE-D-19-15766R2

Dear Dr. Ostman,

We are pleased to inform you that your manuscript has been judged scientifically suitable for publication and will be formally accepted for publication once it complies with all outstanding technical requirements.

With kind regards,

Maciej S. Buchowski

Academic Editor

PLOS ONE

Additional Editor Comments (optional):

Reviewers' comments:

Reviewer's Responses to Questions

**Comments to the Author**

1. If the authors have adequately addressed your comments raised in a previous round of review and you feel that this manuscript is now acceptable for publication, you may indicate that here to bypass the “Comments to the Author” section, enter your conflict of interest statement in the “Confidential to Editor” section, and submit your "Accept" recommendation.

Reviewer #1: All comments have been addressed

Reviewer #2: All comments have been addressed

Reviewer #3: All comments have been addressed

2. Is the manuscript technically sound, and do the data support the conclusions?

Reviewer #1: (No Response)

Reviewer #2: Yes

Reviewer #3: (No Response)

3. Has the statistical analysis been performed appropriately and rigorously? 

Reviewer #1: (No Response)

Reviewer #2: Yes

Reviewer #3: (No Response)

4. Have the authors made all data underlying the findings in their manuscript fully available?

Reviewer #1: (No Response)

Reviewer #2: No

Reviewer #3: (No Response)

5. Is the manuscript presented in an intelligible fashion and written in standard English?

Reviewer #1: (No Response)

Reviewer #2: Yes

Reviewer #3: (No Response)

6. Review Comments to the Author

Reviewer #1: Line 135 - the sentence that has been added since the last review does not add any clarity, if anything it makes the description harder to understand. I was hoping for a more 'lay-appropriate' explanation for what the incremental area-under-the-curve represents. I had previously tempted the authors to add clarity by asking them "Does it represent a change in plasma glucose concentration over 120 minutes post-ingestion of the product?" I was hoping that in answering this one way or another, the authors would have been able to explain what this outcome measure represents. This is a missed opportunity, however, I am not inclined to hold up the publication of this excellent piece of research on this account!

Reviewer #2: (No Response)

Reviewer #3: (No Response)

7. PLOS authors have the option to publish the peer review history of their article (what does this mean?). If published, this will include your full peer review and any attached files.

Reviewer #1: No

Reviewer #2: No

Reviewer #3: Yes: Jose Donato Jr.

---

## [Editor Report · Acceptance letter]

29 May 2020

PONE-D-19-15766R2 

A novel nutritional supplement containing amino acids and chromium decreases postprandial glucose response in a randomized, double-blind, placebo-controlled study 

Dear Dr. Östman:

I am pleased to inform you that your manuscript has been deemed suitable for publication in PLOS ONE. Congratulations! Your manuscript is now with our production department. 

With kind regards,

on behalf of

Dr. Maciej S. Buchowski 

Academic Editor

PLOS ONE